# A Novel Second-Order Neurodynamic System to Fixed-Time Nash Equilibrium Seeking

Yao Song[1], Xingxing Ju[1,2], Chaoli Yao[3]

1, College of Electronics and Information Engineering, Sichuan University, Chengdu 610065, China

2, Key Laboratory of System Control and Information Processing, Ministry of Education,
Shanghai Jiaotong University, Shanghai 200240, China

3, Key Laboratory of Engineering Modeling and Statistical Computation of Hainan Province,
College of Mathematics and Statistics, Hainan University, Hainan 570228, China

Email: songyao@stu.scu.edu.cn, xingxju@scu.edu.cn, clyao@hainanu.edu.cn

*Abstract*—This paper investigates fixed-time (FT) Nash equilibrium (NE) seeking problem for non-cooperative games. A novel second-order NE seeking neurodynamic system is proposed to accelerate the convergence. It is proved that the proposed neurodynamic system's trajectory converges to NE within fixed-time under a function called potential function and strongly monotone potential function respectively. It is shown that the upper bounds are independent of the initial conditions. The robustness of the proposed NE seeking neurodynamic system under bounded perturbations is further studied. The effectiveness and practicability of the proposed NE seeking system are illustrated via a simulation example and an analog circuit implementation on Multisim 14.3.

*Index Terms*—Non-cooperative games, Nash Equilibrium, fixed-time, neurodynamic system

## I. INTRODUCTION

Non-cooperative games are invaluable for modeling and analyzing interactive decision-making processes [1], [2]. In non-cooperative games, players aim to minimize their individual cost functions to ultimately achieve a stable state identified as the Nash equilibrium. The study of Nash equilibrium in non-cooperative games has become a crucial area of research with significant potential in recent years, which is particularly evident in applications across economics, resource allocation, optimization problems, and energy control [3]–[5].

Neurodynamic systems based on Lyapunov function are well-developed in NE seeking problems, and have appealed a lot of interest in [6]–[11]. To obtain a faster convergence, a variety of accelerated neurodynamics have been investigated and proposed [12]–[15]. A generalized framework for designing accelerated algorithms with strongest convergence was proposed in [16]. Ye [17] considered the bounded inputs and further proposed the first-order and second-order NE seeking dynamics. As a matter of fact, most of the neurodynamic systems in the literature have only established asymptotic convergence, or exponential convergence [18]–[21] while many practical Nash equilibrium seeking problems are expected to be completed within a finite time rather than an infinite one.

Thus recently finite-time (FIT) convergence and fixed-time (FT) convergence have been investigated frequently. An algorithm was proposed to achieve finite-time Nash equilibrium seeking in [22]. While in finite-time NE seeking systems, NE can be guaranteed to realize in finite-time, the upper bound is highly related to the initial conditions. If the initial information of the system is unknown, the convergence time can not be estimated, which will limit the practical application range of the finite-time NE seeking systems. To address the mentioned limitations of finite-time NE seeking neurodynamic systems, fixed-time NE seeking systems have been hence developed and investigated in [23], [24]. Fixed-time NE seeking system can converge to NE within fixed-time and the upper bound does not depend on the initial conditions of the players. Therefore, investigating fixed-time NE seeking system holds significant research value. To be specific, Li *et al.* [23] investigated a finite-time and a fixed-time NE seeking algorithm within predictable time respectively, which is independent of the initial states. For NE seeking under both constant and time-varying delays, Ai [24] proposed the gradient play technique with FT differentiator. Moreover, the implementations of neurodynamic system have attracted the attention and well developed in recent years. Wu *et al.* [25] presented two novel analog circuit frameworks in presence of a non-smooth term, formed by integrator, gradient estimate, proximal operator and other basic operation models. While the analog circuit implementations of fixed-time NE seeking neurodynamic systems have not been well developed.

Literature review above reveals that most works achieve NE seeking either with asymptotic or exponential convergence and the second-order neurodynamic system have not been well developed in fixed-time NE seeking problems. Motivated by the literature reviews above, in this paper we focus on designing a novel second-order neurodynamic system for fixed-time NE seeking with strong robustness, and its analog circuit implementation. The main contributions and innovations are summarized as follows.

We propose a novel second-order neurodynamic system to achieve FT convergence for NE seeking problems under

This work was supported in part by the China Postdoctoral Science Foundation under Grant 2023M742457, in part by the Foundation of Key Laboratory of System Control and Information Processing of Ministry of Education of China under Grant Scip20240107, and in part by the Key Laboratory of Engineering Modeling and Statistical Computation of Hainan Province under Grant HNGCTJ2407.

strongly monotone function and potential function respectively. The upper bounds are explicitly given, which are independent of the initial conditions of the players. Then, we show that the proposed FT convergence of the neurodynamic system is preserved when it is subject to some bounded noise. To validate the physical realizability of designed system, we implement the proposed NE seeking neurodynamic system on the analog circuit.

The rest of the paper is organized as follows. In Section II, the problem formulation and some lemmas are given. In Section III, a novel second-order neurodynamic system for fixed-time NE seeking is proposed as well as its convergence and robustness analysis. Further, in Section IV, a numerical simulation and circuit experiment are provided to illustrate effectiveness and advantages of the proposed system, which is followed by some concluding remarks in Section V.

**Notations**: The paper adopts $\mathbb{R}$, $\mathbb{R}^n$ to denote the set of real numbers and vectors respectively. $\|\cdot\|$ and $|x|$ denotes 2-norm and component-wise absolute value of the vector $x \in \mathbb{R}^n$ respectively.

## II. PRELIMINARIES

### A. Problem fomulation

Consider $N$ players in the non-cooperative game with two or more participants. The set of players is denoted as : $\mathcal{N} = \{1, 2, \ldots, N\} \in \mathbb{R}^N$. The strategy function that consists of all the players' actions is described as $x = [x_1, x_2, \ldots, x_N] \in \mathbb{R}^N$, where $x_i$ represents the actions that player $i$ can take. $x_{-i} = [x_1, \ldots, x_{i-1}, x_{i+1}, \ldots, x_N] \in \mathbb{R}^{N-1}$ denotes the function consisting of all other players' actions except $x_i$. Thus, $x$ can be represented as $x = (x_i, x_{-i})$. The cost function for player $i$ is denoted as $f_i$, representing the cost for each player. When the strategy profile $x^*$ satisfies the NE condition, the cost is minimized for each player.

For notation convenience, we get that $f_i = f_i(x_i, x_{-i})$. Then the pseudogradient is defined as: $G(x) = [\nabla_1 f_1(x), \nabla_2 f_2(x), \ldots, \nabla_N f_N(x)]^\top$, where each element $\nabla_i f_i$ is defined as $\nabla_i f_i(x) = \frac{\partial f_i(x)}{\partial x_i}$, for all $i \in \mathcal{N}$. And we define $r(x)$ as

$$r(x) = \begin{bmatrix} \frac{\partial^2 f_1(x)}{\partial^2 x_1} & \frac{\partial^2 f_1(x)}{\partial x_1 \partial x_2} & \cdot & \cdot & \cdot & \frac{\partial^2 f_1(x)}{\partial x_1 \partial x_N} \\ \frac{\partial^2 f_2(x)}{\partial x_2 \partial x_1} & \frac{\partial^2 f_2(x)}{\partial^2 x_2} & \cdot & \cdot & \cdot & \frac{\partial^2 f_2(x)}{\partial x_2 \partial x_N} \\ \cdot & \cdot & \cdot & \cdot & \cdot & \cdot \\ \cdot & \cdot & \cdot & \cdot & \cdot & \cdot \\ \frac{\partial^2 f_N(x)}{\partial x_N \partial x_1} & \frac{\partial^2 f_N(x)}{\partial x_N \partial x_2} & \cdot & \cdot & \cdot & \frac{\partial^2 f_2(x)}{\partial^2 x_N} \end{bmatrix}.$$

The NE for a non-cooperative game is defined as $x^* = (x_i^*, x_{-i}^*)$ such that

$$f_i(x_i^*, x_{-i}^*) \leq f_i(x_i, x_{-i}^*). \tag{1}$$

NE means that no player can further reduce its associated cost function by unilaterally changing its own action.

### B. Lemmas

**Lemma 1.** *[26]* Consider the general differential equation

$$\dot{x} = f(x(t)), x(0) = x_0 \in \mathbb{R}^n, \tag{2}$$

where $x \in \mathbb{R}^n$ and $f : \mathbb{R} \times \mathbb{R}^n \to \mathbb{R}^n$ is a time-continuous non-linear function, there exists a $x^* \in \mathbb{R}^n$ is an equilibrium point of system (1) such that $f(x^*) = 0$ for all $t \geq 0$.

Consider system (2). If there exists a continuous radially unbounded function $V : \mathbb{R}^n \to \mathbb{R}_+ \bigcup \{0\}$ such that

- $V(x(t)) = 0 \Leftrightarrow x(t) = 0$.
- any solution $x(t)$ of (2) satisfies the inequality

$$\dot{V}(x(t)) \leq -\mu V(x(t)) - \alpha^k V^{pk}(x(t)) - \beta^k V^{qk}(x(t))$$

for some $\mu, \alpha, \beta, p, q, k > 0 : pk < 1, qk > 1$.

Then the origin of the system (2) is globally fixed-time stable, and the following estimate of the settling time $T(x_0)$ holds

$$T(x_0) \leq \hat{T} = \frac{ln(1 + \frac{\mu}{\alpha^k})}{\mu(1 - pk)} + \frac{ln(1 + \frac{\mu}{\beta^k})}{\mu(qk - 1)}. \tag{3}$$

**Lemma 2.** [27] Let $\xi_i \geq 0$ for all $i \in \mathcal{N}$ and $a, \beta > 1$, it holds that,

$$(\sum_{i=1}^N \xi_i)^{\frac{1}{a}} \leq \sum_{i=1}^N \xi_i^{\frac{1}{a}}, N^{1-\beta}(\sum_{i=1}^N \xi_i)^\beta \leq \sum_{i=1}^N \xi_i^\beta. \tag{4}$$

Note that if the unbounded function $V$ satisfies the inequality in Lemma 1, the estimation of the settling time bound can be achieved as $\hat{T}$. Therefore, the estimation of the settling time bound is independent of the initial state $x(0)$.

## III. MAIN RESULTS

### A. FT Nash equilibrium seeking neurodynamic system

To achieve fixed-time convergence for NE seeking, we consider the following second-order neurodynamic system,

$$\dot{z}_i = \nabla_i f_i(x) - z_i \left( \frac{m_i}{\|z_i\|^a} + \frac{n_i}{\|z_i\|^{-b}} + l_i \right), \tag{5a}$$

$$\dot{x}_i = -z_i - \nabla_i f_i(x) \left( \frac{g_i}{\|\nabla_i f_i(x)\|^a} + \frac{q_i}{\|\nabla_i f_i(x)\|^{-b}} + h_i \right), \tag{5b}$$

where $m_i, n_i, g_i, q_i > 0$ are tunable gains, $a \in (0, 1)$ and $b > 0$ are tunable parameters.

The diagram framework of neurodynamic system (5) is simply depicted in Fig. 1 with basic electronic components including operational amplifier, resistor, capacitor and a bunch of other components.

To achieve fixed-time convergence of neurodynamic system (5), we make the following assumptions.

**Assumption 1.** The pesudogradient $G(x)$ is strongly monotone and Lipschitz continuous with $\alpha > 0$ that $\alpha\|x - y\|^2 \leq \langle G(x) - G(y), x - y \rangle$.

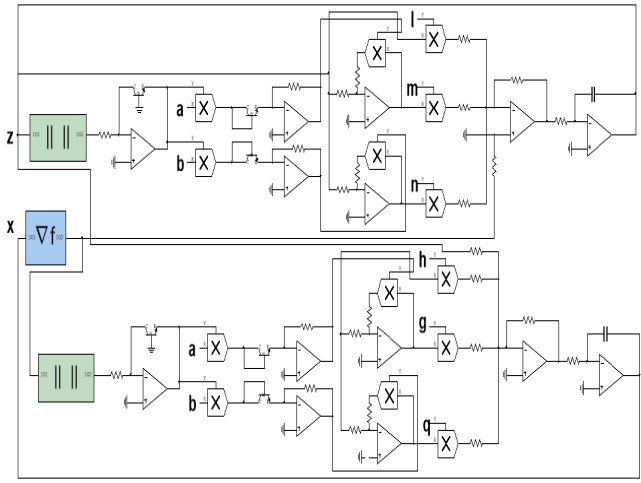

Fig. 1: Circuit frame of neurodynamic system (5).

**Assumption 2.** There exists a unbounded function $F$ and $w > 0$. And the function $F$ satisfies the follows equalities and inequality, which is called potential function,

$$\nabla_i F(x) = \nabla_i f_i(x), \tag{6}$$

$$x^* = \arg_{\min} F(x), \tag{7}$$

$$F(x) - F(x^*) \le \frac{1}{2w} \|G(x)\|^2. \tag{8}$$

**Remark 1.** Under Assumption 1, $r(x)$ satisfies $r(x) + r(x)^\top \ge 2\lambda I$, $I$ is identity matrix and $x \in \mathbb{R}^n$. Note that any potential game with a strongly convex potential function satisfies Assumption 2. In this condition, we can relax the strongly-convex function. The existence of NE is guaranteed under these assumptions.

**Theorem 1.** Suppose that Assumption 1 holds. Let $g_0 = \min\{g_i\}$, $q_0 = \min\{q_i\}$, $m_0 = \min\{m_i\}$, $n_0 = \min\{n_i\}$, $h_0 = \min\{h_i\}$ and $l_0 = \min\{l_i\}$ for all $i \in \mathcal{N}$. Then the trajectory $x(t)$ of neurodynamic system (5) converges to NE $x^*$ in FT with the settling time given as follows,

$$\hat{T} \le \frac{2ln(1 + \frac{y_3}{y_1})}{y_3 a} + \frac{2ln(1 + \frac{y_3}{y_2})}{y_3 b}, \tag{9}$$

where $y_1 = \min\{2^{1-\frac{a}{2}}\lambda g_0, (2\lambda)^{1-\frac{a}{2}}m_0\}$, $y_2 = \min\{2^{1+\frac{b}{2}}\lambda q_0 N^{-b}, (2\lambda)^{1+\frac{b}{2}}n_0 N^{-b}\}$, $y_3 = \min\{2\lambda h_0, 2\lambda l_0\}$ are positive constants.

*Proof.* Consider the Lyapunov function

$$V(x) = \frac{1}{2}\|G(x)\|^2 + \frac{\lambda}{2}\sum_{i=1}^{N} z_i^2, \tag{10}$$

which is positive definite with respect to $x^*$, and also radially unbounded. The time derivative of $V(x)$ satisfies

$$\dot{V}(x) = -[\nabla_i f_i(x)]_{\text{vec}}^\top r(x)[\frac{g_i \nabla_i f_i(x)}{\|\nabla_i f_i(x)\|^a}]_{\text{vec}}$$
$$- [\nabla_i f_i(x)]_{\text{vec}}^\top r(x)[\frac{q_i \nabla_i f_i(x)}{\|\nabla_i f_i(x)\|^b}]_{\text{vec}}$$
$$- [\nabla_i f_i(x)]_{\text{vec}}^\top r(x)[h_i \nabla_i f_i(x)]_{\text{vec}}$$
$$- \lambda m_i \sum_{i=1}^{N} z_i^{2-a} - \lambda n_i \sum_{i=1}^{N} z_i^{2+b} - \lambda l_i \sum_{i=1}^{N} z_i^2.$$

It follows from Lemma 2 that

$$\dot{V}(x) \le -\lambda[\nabla_i f_i(x)]_{\text{vec}}^\top I(x)[\frac{g_i \nabla_i f_i(x)}{\|\nabla_i f_i(x)\|^a}]_{\text{vec}}$$
$$- \lambda[\nabla_i f_i(x)]_{\text{vec}}^\top I(x)[\frac{q_i \nabla_i f_i(x)}{\|\nabla_i f_i(x)\|^b}]_{\text{vec}}$$
$$- \lambda[\nabla_i f_i(x)]_{\text{vec}}^\top I(x)[h_i \nabla_i f_i(x)]_{\text{vec}}$$
$$- \lambda m_0 \sum_{i=1}^{N} z_i^{2-a} - \lambda n_0 \sum_{i=1}^{N} z_i^{2+b} - \lambda l_i \sum_{i=1}^{N} z_i^2$$
$$\le -\lambda g_0 \left(\sum_{i=1}^{N} \|\nabla_i f_i(x)\|^2\right)^{1-\frac{a}{2}}$$
$$- \lambda q_0 N^{-\frac{b}{2}} \left(\sum_{i=1}^{N} \|\nabla_i f_i(x)\|^2\right)^{1+\frac{b}{2}}$$
$$- \lambda h_0 \sum_{i=1}^{N} \|\nabla_i f_i(x)\|^2 - \lambda m_0 \left(\sum_{i=1}^{N} z_i^2\right)^{1-\frac{a}{2}}$$
$$- \lambda n_0 N^{-\frac{b}{2}} \left(\sum_{i=1}^{N} z_i^2\right)^{1+\frac{b}{2}} - \lambda l_i \sum_{i=1}^{N} z_i^2.$$

Under Assumption 1, we get

$$\dot{V}(x) \le -2^{1-\frac{a}{2}}\lambda g_0 \left(\frac{1}{2}\|G(x)\|^2\right)^{1-\frac{a}{2}}$$
$$- 2^{1+\frac{b}{2}}\lambda q_0 N^{-\frac{b}{2}} \left(\frac{1}{2}\|G(x)\|^2\right)^{1+\frac{b}{2}}$$
$$- 2\lambda h_0 \left(\frac{1}{2}\|G(x)\|^2\right) - (2\lambda)^{1-\frac{a}{2}}m_0 \left(\frac{\lambda}{2}\sum_{i=1}^{N} z_i^2\right)^{1-\frac{a}{2}}$$
$$- (2\lambda)^{1+\frac{b}{2}}n_0 N^{-\frac{b}{2}} \left(\frac{\lambda}{2}\sum_{i=1}^{N} z_i^2\right)^{1+\frac{b}{2}} - 2\lambda l_0 \left(\frac{\lambda}{2}\sum_{i=1}^{N} z_i^2\right).$$

Combining the Lyapunov function in eq. (10), we can further express the time derivative of $V(x)$ as follows

$$\dot{V}(x) \le -y_1 V(x)^{1-\frac{a}{2}} - y_2 V(x)^{1+\frac{b}{2}} - y_3 V(x),$$

where the exponential terms $(1 - \frac{a}{2}) < 1$, $(1 + \frac{b}{2}) > 1$ and $y_1 = \min\{2^{1-\frac{a}{2}}\lambda g_0, (2\lambda)^{1-\frac{a}{2}}m_0\} > 0$, $y_2 = \min\{2^{1+\frac{b}{2}}\lambda q_0 N^{-b}, (2\lambda)^{1+\frac{b}{2}}n_0 N^{-b}\} > 0$, $y_3 = \min\{2\lambda h_0, 2\lambda l_0\} > 0$. Combining Lemma 1, the settling time can be easily obtained as (9). This also implies that $x^*$ is an equilibrium point of the neurodynamic. The proof is thus completed. $\square$

**Remark 2.** Note that novel bound inequality (9) depends on the designed parameters $a, b, g_i, m_i, n_i, h_i, q_i$ and the number of players $N$, which can all be pre-set. Thus, the settling time bound $\hat{T} > 0$ can calculated via the assignment of the gains above.

We now consider the neurodynamic system (5) with the potential function as shown in Assumption 2.

**Theorem 2.** Suppose that Assumption 2 holds. Let $g_0 = \min\{g_i\}$, $q_0 = \min\{q_i\}$, $m_0 = \min\{m_i\}$, $n_0 = \min\{n_i\}$, $h_0 = \min\{h_i\}$ and $l_0 = \min\{l_i\}$ for all $i \in \mathcal{N}$. Then the trajectory $x(t)$ of neurodynamic system (5) converges to NE $x^*$ in FT with the settling time given as follows,

$$\hat{T} \leq \frac{2ln(1 + \frac{u_3}{u_1})}{u_3 a} + \frac{2ln(1 + \frac{u_3}{u_2})}{u_3 b}, \quad (11)$$

where $u_1 = \min\{(2w)^{1-\frac{a}{2}}g_0, 2^{1-\frac{a}{2}}m_0\}$, $u_2 = \min\{(2w)^{1+\frac{b}{2}}q_0 N^{-b}, 2^{1+\frac{b}{2}}n_0 N^{-b}\}$, $u_3 = \min\{2wh_0, 2l_0\}$ are positive constants.

*Proof.* Consider the Lyapunov function

$$V(x) = F(x) - F(x^*) + \frac{1}{2}\sum_{i=1}^{n} z_i^2, \quad (12)$$

which is also positive definite with respect to $x^*$, and also radially unbounded due to Assumption 2. The time derivative of $V(x)$ satisfies

$$\dot{V}(x) = \sum_{i=1}^{N} \nabla_i f_i(x) \dot{x}_i + \sum_{i=1}^{N} z_i \dot{z}_i. \quad (13)$$

It follows from Lemma 2 that

$$\dot{V}(x) = g_i \sum_{i=1}^{N}\left(\|\nabla_i f_i(x)\|^2\right)^{1-\frac{a}{2}} - q_i \sum_{i=1}^{N}\left(\|\nabla_i f_i(x)\|^2\right)^{1+\frac{b}{2}}$$
$$- h_0 \sum_{i=1}^{N} \|\nabla_i f_i(x)\|^2 - m_i \sum_{i=1}^{N}\left(z_i^2\right)^{1-\frac{a}{2}}$$
$$- n_i \sum_{i=1}^{N}\left(z_i^2\right)^{1+\frac{b}{2}} - l_i \sum_{i=1}^{N} z_i^2$$
$$\leq -g_0\left(\sum_{i=1}^{N} \|\nabla_i f_i(x)\|^2\right)^{1-\frac{a}{2}}$$
$$- q_0 N^{-\frac{b}{2}}\left(\sum_{i=1}^{N} \|\nabla_i f_i(x)\|\right)^{1+\frac{b}{2}}$$
$$- h_0 \sum_{i=1}^{N} \|\nabla_i f_i(x)\| - m_0\left(\sum_{i=1}^{N} z_i^2\right)^{1-\frac{a}{2}}$$
$$- n_0 N^{-\frac{b}{2}}\left(\sum_{i=1}^{N} z_i^2\right)^{1+\frac{b}{2}} - l_i \sum_{i=1}^{N} z_i^2.$$

Using the property of Assumption 2, we can get

$$\dot{V}(x) \leq -g_0 \|G(x)\|^{2-a} - q_0 N^{-\frac{b}{2}}\|G(x)\|^{2+b}$$
$$- h_0 \|G(x)\|^2 - m_0\left(\sum_{i=1}^{N} z_i^2\right)^{1-\frac{a}{2}}$$
$$- n_0 N^{-\frac{b}{2}}\left(\sum_{i=1}^{N} z_i^2\right)^{1+\frac{b}{2}} - l_0 \sum_{i=1}^{N} z_i^2$$
$$\leq -(2w)^{1-\frac{a}{2}}g_0\left(F(x) - F(x^*)\right)^{1-\frac{a}{2}}$$
$$- (2w)^{1+\frac{b}{2}}q_0 N^{-\frac{b}{2}}\left(F(x) - F(x^*)\right)^{1+\frac{b}{2}}$$
$$- 2wh_0\left(F(x) - F(x^*)\right) - m_0\left(\sum_{i=1}^{N} z_i^2\right)^{1-\frac{a}{2}}$$
$$- n_0 N^{-\frac{b}{2}}\left(\sum_{i=1}^{N} z_i^2\right)^{1+\frac{b}{2}} - l_0 \sum_{i=1}^{N} z_i^2.$$

Combining the Lyapunov function in eq. (12), we can further express the upper bound of the time derivative of $V(x)$ as follows,

$$\dot{V}(x) \leq -u_1 V(x)^{1-\frac{a}{2}} - u_2 V(x)^{1+\frac{b}{2}} - u_3 V(x),$$

where the exponential terms $(1 - \frac{a}{2}) < 1$, $(1 + \frac{b}{2}) > 1$ and $u_1 = \min\{(2w)^{1-\frac{a}{2}}g_0, 2^{1-\frac{a}{2}}m_0\} > 0$, $u_2 = \min\{(2w)^{1+\frac{b}{2}}q_0 N^{-b}, 2^{1+\frac{b}{2}}n_0 N^{-b}\} > 0$, $u_3 = \min\{2wh_0, 2l_0\} > 0$. It follows from Lemma 1 that the trajectory $x(t)$ of system (5) converges to NE $x^*$ in FT with the settling time given as in (11). It implies that $x^*$ is an equilibrium point of the neurodynamic system. The proof is thus completed. $\square$

**Remark 3.** It can be observed that the proposed neurodynamic system either under Assumption 1 or Assumption 2 can guarantee the existence and uniqueness of NE. Moreover, different from the literature work with asymptotic or exponential convergence [18]–[21], we propose accelerated and explicit fixed-time upper bounds.

*B. Robustness analysis*

We now investigate the robustness of neurodynamic system (5), the FT convergence is maintained when it is exposed to the following perturbations,

$$\dot{z}_i = \nabla_i f_i(x) - z_i\left(\frac{m_i}{\|z_i\|^a} + \frac{n_i}{\|z_i\|^{-b}} + l_i\right) + \zeta_i(z), \quad (14a)$$

$$\dot{x}_i = z_i - \nabla_i f_i(x)\left(\frac{g_i}{\|\nabla_i f_i(x)\|^a} + \frac{q_i}{\|\nabla_i f_i(x)\|^{-b}}\right.$$
$$\left. + h_i\right) + \omega_i(x), \quad (14b)$$

where $\zeta_i(z), \omega_i(x) : \mathbb{R}^n \to \mathbb{R}$ are two perturbation terms, satisfy the following assumption.

**Assumption 3.** There exist constants $L_1 > 0$ and $L_2 > 0$ such that those perturbations above $\|\zeta_i(z)\| < L_1\|z\|$ and $\|\omega_i(x)\| < L_2\|\nabla_i f_i(x)\|$ for all $i \in \mathcal{N}$.

**Theorem 3.** Suppose Assumption 1 and Assumption 3 hold. Let $h_0 \geq L_1$, $l_0 \geq L_2$, $g_0 = \min\{g_i\}$, $q_0 = \min\{q_i\}$, $m_0 = \min\{m_i\}$, $n_0 = \min\{n_i\}$, $h_0 = \min\{h_i\}$ and $l_0 = \min\{l_i\}$ for all $i \in \mathcal{N}$. Then the trajectory $x(t)$ of the neurodynamic system (14) converges to NE $x^*$ in FT with the settling time given as follows,

$$\hat{T} \leq \frac{2ln(1 + \frac{y_3^*}{y_1^*})}{y_3^* a} + \frac{2ln(1 + \frac{y_3^*}{y_2^*})}{y_3^* b}, \qquad (15)$$

where $y_1^* = \min\{2^{1-\frac{a}{2}}\lambda g_0, (2\lambda)^{1-\frac{a}{2}}m_0\}$, $y_2^* = \min\{2^{1+\frac{b}{2}}\lambda q_0 N^{-b}, (2\lambda)^{1+\frac{b}{2}}n_0 N^{-b}\}$, $y_3^* = \min\{2\lambda(h_0 - L_1), 2\lambda(l_0 - L_2)\}$ are some positive constants.

*Proof.* Consider the Lyapunov function in eq. (10), the time derivative of $V(x)$ satisfies

$$\begin{aligned}
\dot{V}(x) = & -[\nabla_i f_i(x)]_{\text{vec}}^\top r(x)[\frac{g_i \nabla_i f_i(x)}{\|\nabla_i f_i(x)\|^a}]_{\text{vec}} \\
& -[\nabla_i f_i(x)]_{\text{vec}}^\top r(x)[\frac{q_i \nabla_i f_i(x)}{\|\nabla_i f_i(x)\|^b}]_{\text{vec}} \\
& -[\nabla_i f_i(x)]_{\text{vec}}^\top r(x)[h_i \nabla_i f_i(x)]_{\text{vec}} \\
& +[\nabla_i f_i(x)]_{\text{vec}}^\top r(x)[L_1 \nabla_i f_i(x)]_{\text{vec}} \\
& -\lambda m_i \sum_{i=1}^N z_i^{2-a} - \lambda n_i \sum_{i=1}^N z_i^{2+b} \\
& -\lambda l_i \sum_{i=1}^N z_i^2 + \lambda L_2 \sum_{i=1}^N z_i^2.
\end{aligned}$$

It follows from Lemma 2 that

$$\begin{aligned}
\dot{V}(x) \leq & -\lambda[\nabla_i f_i(x)]_{\text{vec}}^\top I(x)[\frac{g_i \nabla_i f_i(x)}{\|\nabla_i f_i(x)\|^a}]_{\text{vec}} \\
& -\lambda[\nabla_i f_i(x)]_{\text{vec}}^\top I(x)[\frac{q_i \nabla_i f_i(x)}{\|\nabla_i f_i(x)\|^b}]_{\text{vec}} \\
& -\lambda[\nabla_i f_i(x)]_{\text{vec}}^\top I(x)[h_i \nabla_i f_i(x)]_{\text{vec}} \\
& +\lambda[\nabla_i f_i(x)]_{\text{vec}}^\top I(x)[L_1 \nabla_i f_i(x)]_{\text{vec}} \\
& -\lambda m_0 \sum_{i=1}^N z_i^{2-a} - \lambda n_0 \sum_{i=1}^N z_i^{2+b} \\
& -\lambda l_i \sum_{i=1}^N z_i^2 + \lambda L_2 \sum_{i=1}^N z_i^2 \\
\leq & -\lambda g_0 \left(\sum_{i=1}^N \|\nabla_i f_i(x)\|^2\right)^{1-\frac{a}{2}} \\
& -\lambda q_0 N^{-\frac{b}{2}} \left(\sum_{i=1}^N \|\nabla_i f_i(x)\|^2\right)^{1+\frac{b}{2}} \\
& -\lambda(h_0 - L_1)\sum_{i=1}^N \|\nabla_i f_i(x)\|^2 - \lambda m_0 \left(\sum_{i=1}^N z_i^2\right)^{1-\frac{a}{2}} \\
& -\lambda n_0 N^{-\frac{b}{2}}\left(\sum_{i=1}^N z_i^2\right)^{1+\frac{b}{2}} - \lambda(l_0 - L_2)\sum_{i=1}^N z_i^2.
\end{aligned}$$

Combining Assumption 1, we get

$$\begin{aligned}
\dot{V}(x) \leq & -2^{1-\frac{a}{2}}\lambda g_0 \left(\frac{1}{2}\|G(x)\|^2\right)^{1-\frac{a}{2}} \\
& -2^{1+\frac{b}{2}}\lambda q_0 N^{-\frac{b}{2}}\left(\frac{1}{2}\|G(x)\|^2\right)^{1+\frac{b}{2}} \\
& -2\lambda(h_0 - L_1)\left(\frac{1}{2}\|G(x)\|^2\right) \\
& -(2\lambda)^{1-\frac{a}{2}}m_0 \left(\frac{\lambda}{2}\sum_{i=1}^N z_i^2\right)^{1-\frac{a}{2}} \\
& -(2\lambda)^{1+\frac{b}{2}}n_0 N^{-\frac{b}{2}}\left(\frac{\lambda}{2}\sum_{i=1}^N z_i^2\right)^{1+\frac{b}{2}} \\
& -2\lambda(l_0 - L_2)\left(\frac{\lambda}{2}\sum_{i=1}^N z_i^2\right).
\end{aligned}$$

Under Assumption 3, $h_0 - L_1, l_0 - L_2 > 0$ and combine the Lyapunov function in eq. (10), we can get

$$\dot{V}(x) \leq -y_1^* V(x)^{1-\frac{a}{2}} - y_2^* V(x)^{1+\frac{b}{2}} - y_3^* V(x),$$

where the exponential terms $(1 - \frac{a}{2}) < 1$, $(1 + \frac{b}{2}) > 1$ and $y_1^* = \min\{2^{1-\frac{a}{2}}\lambda g_0, (2\lambda)^{1-\frac{a}{2}}m_0\} > 0$, $y_2^* = \min\{2^{1+\frac{b}{2}}\lambda q_0 N^{-b}, (2\lambda)^{1+\frac{b}{2}}n_0 N^{-b}\} > 0$, $y_3^* = \min\{2\lambda(h_0 - L_1), 2\lambda(l_0 - L_2)\} > 0$. It follows from Lemma 1 that the trajectory $x(t)$ of neurodynamic system (14) converges to NE $x^*$ in FT with the settling time given as in (15). It implies that $x^*$ is an equilibrium point of the neurodynamic system when it is subject to the above perturbations. The proof is thus completed. $\qquad \square$

**Theorem 4.** Suppose Assumption 2 and Assumption 3 hold. Let that $h_0 \geq L_1$, $l_0 \geq L_2$, $g_0 = \min\{g_i\}$, $q_0 = \min\{q_i\}$, $m_0 = \min\{m_i\}$, $n_0 = \min\{n_i\}$, $h_0 = \min\{h_i\}$ and $l_0 = \min\{l_i\}$ for all $i \in \mathcal{N}$. Then the trajectory $x(t)$ of system (14) converges to NE $x^*$ in FT with the settling time given as follows,

$$\hat{T} \leq \frac{2ln(1 + \frac{u_3^*}{u_1^*})}{u_3^* a} + \frac{2ln(1 + \frac{u_3^*}{u_2^*})}{u_3^* b}, \qquad (16)$$

where $u_1^* = \min\{(2w)^{1-\frac{a}{2}}g_0, 2^{1-\frac{a}{2}}m_0\}$, $u_2^* = \min\{(2w)^{1+\frac{b}{2}}q_0 N^{-b}, 2^{1+\frac{b}{2}}n_0 N^{-b}\}$, $u_3^* = \min\{2w(h_0 - L_1), 2(l_0 - L_2)\}$ are some positive constants.

*Proof.* Take the Lyapunov function in eq. (12). Combining Lemma 2, by differentiating $V(x)$ with respect to time, we

get

$$\dot{V}(x) = -g_i \sum_{i=1}^{N} \left( \|\nabla_i f_i(x)\|^2 \right)_i^{1-\frac{a}{2}} - q_i \sum_{i=1}^{N} \left( \|\nabla_i f_i(x)\|^2 \right)_i^{1+\frac{b}{2}}$$

$$- h_0 \sum_{i=1}^{N} \|\nabla_i f_i(x)\|_i^2 + L_1 \sum_{i=1}^{N} \|\nabla_i f_i(x)\|^2$$

$$- m_i \sum_{i=1}^{N} \left( z_i^2 \right)^{1-\frac{a}{2}} - n_i \sum_{i=1}^{N} \left( z_i^2 \right)^{1+\frac{b}{2}}$$

$$- l_i \sum_{i=1}^{N} z_i^2 + L_2 \sum_{i=1}^{N} z_i^2$$

$$\leq -g_0 \left( \sum_{i=1}^{N} \|\nabla_i f_i(x)\|^2 \right)^{1-\frac{a}{2}} + L_1 \sum_{i=1}^{N} \|\nabla_i f_i(x)\|^2$$

$$- h_0 \sum_{i=1}^{N} \|\nabla_i f_i(x)\|^2 - q_0 N^{-\frac{b}{2}} \left( \sum_{i=1}^{N} \|\nabla_i f_i(x)\|^2 \right)^{1+\frac{b}{2}}$$

$$- m_0 \left( \sum_{i=1}^{N} z_i^2 \right)^{1-\frac{a}{2}} - n_0 N^{-\frac{b}{2}} \left( \sum_{i=1}^{N} z_i^2 \right)^{1+\frac{b}{2}}$$

$$- l_i \sum_{i=1}^{N} z_i^2 + L_2 \sum_{i=1}^{N} z_i^2.$$

It follows from Assumption 2 that

$$\dot{V}(x) \leq -g_0 \|G(x)\|^{2-a} - q_0 N^{-\frac{b}{2}} \|G(x)\|^{2+b}$$

$$- (h_0 - L_1) \|G(x)\|^2 - m_0 \left( \sum_{i=1}^{N} z_i^2 \right)^{1-\frac{a}{2}}$$

$$- n_0 N^{-\frac{b}{2}} \left( \sum_{i=1}^{N} z_i^2 \right)^{1+\frac{b}{2}} - (l_0 - L_2) \sum_{i=1}^{N} z_i^2$$

$$\leq -(2w)^{1-\frac{a}{2}} g_0 \left( F(x) - F(x^*) \right)^{1-\frac{a}{2}}$$

$$- (2w)^{1+\frac{b}{2}} q_0 N^{-\frac{b}{2}} \left( F(x) - F(x^*) \right)^{1+\frac{b}{2}}$$

$$- 2w(h_0 - L_1) \left( F(x) - F(x^*) \right) - m_0 \left( \sum_{i=1}^{N} z_i^2 \right)^{1-\frac{a}{2}}$$

$$- n_0 N^{-\frac{b}{2}} \left( \sum_{i=1}^{N} z_i^2 \right)^{1+\frac{b}{2}} - (l_0 - L_2) \sum_{i=1}^{N} z_i^2.$$

Under Assumption 3, $h_0 - L_1, l_0 - L_2 > 0$ and combine the Lyapunov function in eq. (12), we can get

$$\dot{V}(x) \leq -u_1^* V(x)^{1-\frac{a}{2}} - u_2^* V(x)^{1+\frac{b}{2}} - u_3^* V(x),$$

where the exponential terms $(1 - \frac{a}{2}) < 1$, $(1 + \frac{b}{2}) > 1$ and $u_1^* = \min\{(2w)^{1-\frac{a}{2}} g_0, 2^{1-\frac{a}{2}} m_0\} > 0$, $u_2^* = \min\{(2w)^{1+\frac{b}{2}} q_0 N^{-b}, 2^{1+\frac{b}{2}} n_0 N^{-b}\} > 0$, $u_3^* = \min\{2w(h_0 - L_1), 2(l_0 - L_2)\} > 0$. It follows from Lemma 1 that the trajectory $x(t)$ of the neurodynamic system (14) converges to NE $x^*$ in FT with the settling time given as in (16) when it is subject to perturbations above. The proof is completed. $\square$

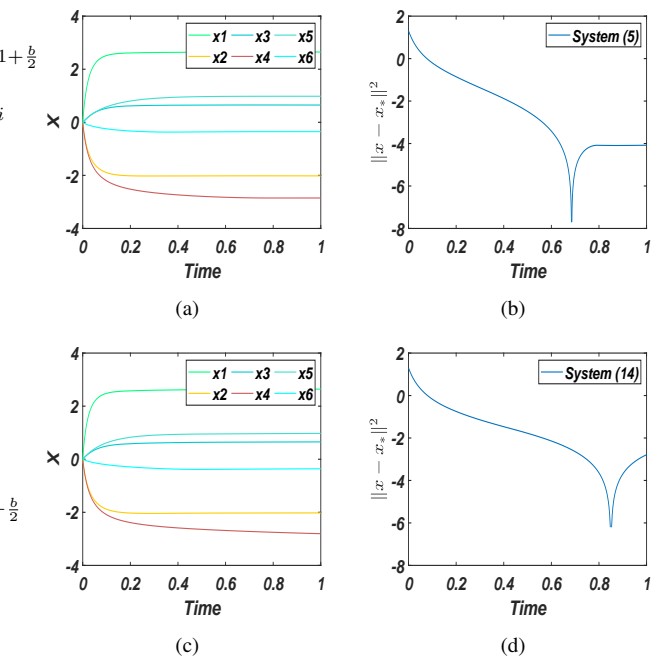

(a)        (b)

(c)        (d)

Fig. 2: (a) Transient responses of neurodynamic system (5); (b) Convergence error responses of neurodynamic system (5) (c) Transient responses of neurodynamic system (14); (d) Convergence error responses of neurodynamic system (14);

## IV. NUMERICAL SIMULATION AND CIRCUIT EXPERIMENT

In this section, We validate the effectiveness of the proposed second-order NE seeking neurodynamic system according to a duopoly game by numerical simulation and analog circuit implementation.

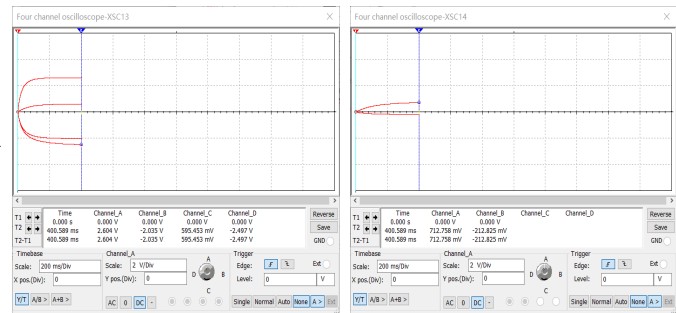

Fig. 4: Circuit implementation results.

We consider $N(N = 6)$ firms producing the same products in a duopoly market. The game is played under the assumption that $N$ players involved in the game are trying to seek NE $x^* = (x_i^*, x_{-i}^*)$ such that

$$f_i(x_i^*, x_{-i}^*) \leq f_i(x_i, x_{-i}^*),$$

where the profit of each company is expressed as $f_i(x_i, x_{-i}) = p_i(x_i - q_i)^2 + (c \sum_{j=1}^{N} x_j + d)x_i$, with $c \sum_{j=1}^{N} x_j + d$ being the pricing function.

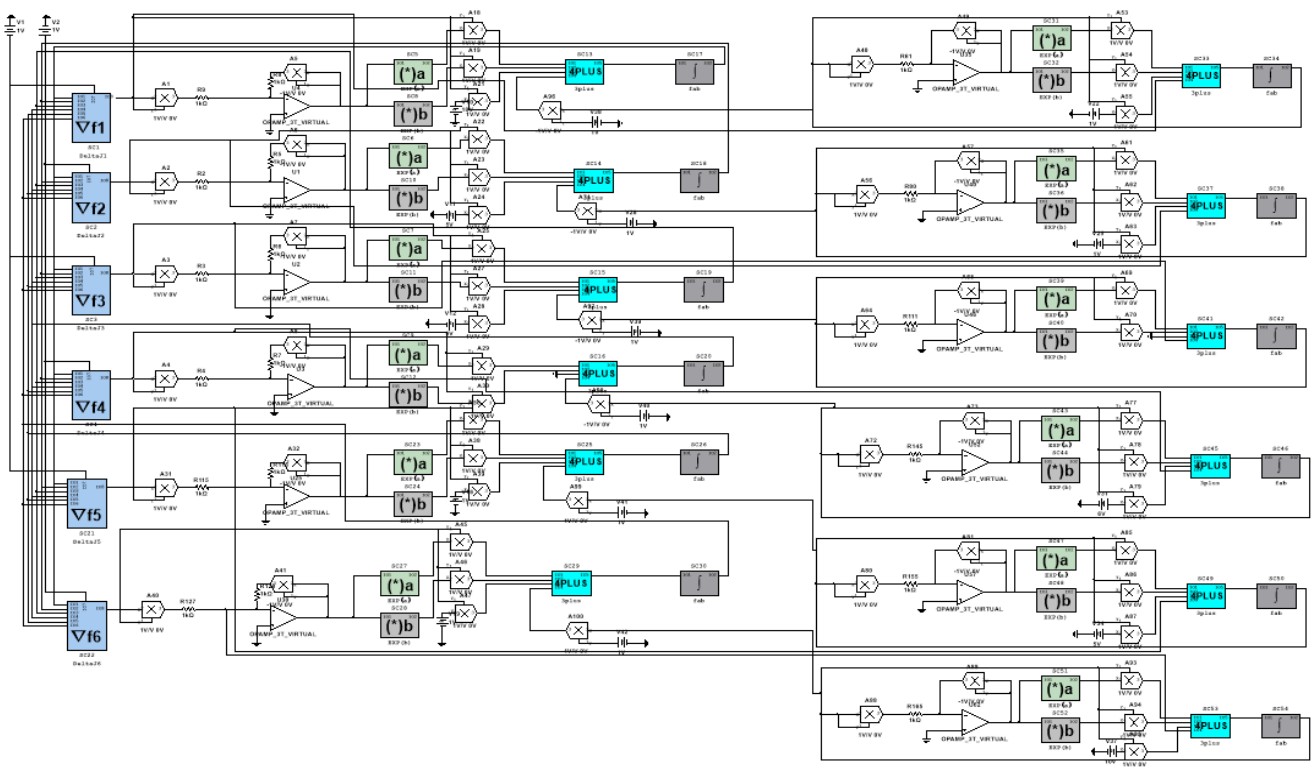

Fig. 3: Circuit implementation of neurodynamic system (5).

TABLE I: **Parameters and results of the experiment**

| Player $i$ | $q_i$ | $h_i$ | $l_i$ | $x_i^*$ in numerical simulation | $x_i^*$ in circuit experiment |
|---|---|---|---|---|---|
| 1 | 8.5 | 10 | 1 | 2.64 | 2.60 |
| 2 | 1.5 | 5 | 1 | -2.0 | -2.03 |
| 3 | 5.5 | 6 | 0.1 | 0.65 | 0.595 |
| 4 | 0.25 | 0.1 | 6 | -2.55 | -2.49 |
| 5 | 6 | 1 | 5 | 0.78 | 0.71 |
| 6 | 4 | 1 | 10 | -0.3 | -0.21 |

Without losing generality, for the proposed second-order NE seeking neurodynamic system, we consider a special case that we suppose $c = p_i = 1$ and $d = 10$, $g_i = q_i = m_i = n_i = 1$ for all $i \in N$ and $a = 0.5$, $b = 1$. The detailed parameters and results achieved by the numerical simulation are shown in Table I.

Fig. 2 (a) shows the changes in participants' behavior in neurodynamic system (5). Fig. 2 (b) indicates that the deviation of participants' actual behavior from the Nash Equilibrium can significantly converges within 0.8 seconds.

It can be seen that participants' behavior quickly approaches the NE under neurodynamic system (14), which has been demonstrated that the neurodynamic system exhibits good robustness. Fig. 2 (d) indicates that the deviation of participants' actual behavior from the Nash Equilibrium can significantly converge within 1 seconds.

The analog circuit framework is designed firstly. The circuit framework is shown in Fig. 1, which contains multipliers, operational amplifiers, capacitors, and resistors as well as a power supply and some other electronics. Fig. 3 shows the whole analog circuit with modularised devices and the results achieved by analog circuits are also shown in Table I.

In Fig. 3, the output of $\nabla f_i$ module is the derivative of $f_i$. After the modules for two-norm and exponentiation, we can get that $\frac{g_i}{\|\nabla_i f_i(x)\|^a}$, $\frac{q_i}{\|\nabla_i f_i(x)\|^{-b}}$, $\frac{m_i}{\|z_i\|^a}$ and $\frac{n_i}{\|z_i\|^{-b}}$. Then after the adder and integrator, we can get the neurodynamic system as in (5).

By analyzing the experimental results, it can be concluded that the proposed neurodynamic system (5) can converge to NE point $x^*$ quickly and remain the stable values, which are almost the same as the numerical simulation calculations. The fixed-time convergence is guaranteed and once the parameters of the neurodynamic system are given, the explicit upper bound can be estimated.

## V. Conclusion

This paper proposes a novel second-order NE seeking neurodynamic system. It has been proved that the fixed-time convergence of the proposed neurodynamic system is independent of the initial states of the players. The explicit upper bound can be calculated according to the designed parameters and the circuit implementation can basically satisfy the practical requirements of applications. Future work will be focused on studying the accelerated distributed NE seeking neurodynamic system.

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
