# OpenReview forum: "A Novel Second-Order Neurodynamic System to Fixed-Time Nash Equilibrium Seeking"
_IEEE.org/ICIST/2024/Conference — IEEE ICIST 2024 Conference Submission_

### Official Review · Reviewer_odja · 2024-08-21
**This paper investigates fixed-time Nash equilibrium seeking problem for non-cooperative games. A  novel second-order NE seeking neurodynamic system is proposed  to accelerate the convergence. The referee thinks it contains some publishable materials and it is worthy of publishing after some revisions.**

**Rating:** 7
**Confidence:** 3

**Review:**

1. There are some grammatical mistakes and typos. Please examine the full text further and revise them.
2. It can be compared with existing articles to make the innovation point clearer.
3. In the simulation section, the clarity of the figure is too low, it is recommended to convert the image to '.eps ' format.

---

### Official Review · Reviewer_JtYo · 2024-08-21
**accept**

**Rating:** 7
**Confidence:** 3

**Review:**

Comment: This paper investigates fixed-time (FT) Nash equilibrium (NE) seeking problem for non-cooperative games. A novel second-order NE seeking neurodynamic system is proposed to accelerate the convergence. The theory is correct and can be accepted after responding the following comments.
(1) More comprehensive literature review is needed to clarify the research gap and research motivation.
(2) The format of several paragraphs in the article is inaccurate, and there is no first line indentation.
(3) The conclusion of the article suggests using the present perfect tense for description.

---

### Official Review · Reviewer_hm2t · 2024-08-26
**A Novel Second-Order Neurodynamic System to Fixed-Time Nash Equilibrium Seeking**

**Rating:** 7
**Confidence:** 2

**Review:**

This paper investigates fixed-time (FT) Nash equilibrium (NE) seeking problem for non-cooperative games. The obtained result is valuable and can be accepted if the following problems can be clarified.
1. The paper should include comparisons against the existing literature to demonstrate its advantages.
2. How does this article deal with actuator fault?
3. The related references should be added to the Assumptions to show the rationality of them.

---

### Decision · Program_Chairs · 2024-09-06

Accept (Oral)